# Genomic signatures suggesting adaptation to ocean acidification in a coral holobiont from volcanic CO$_2$ seeps

Carlos Leiva [1✉], Rocío Pérez-Portela[2,3] & Sarah Lemer[1]

Ocean acidification, caused by anthropogenic CO$_2$ emissions, is predicted to have major consequences for reef-building corals, jeopardizing the scaffolding of the most biodiverse marine habitats. However, whether corals can adapt to ocean acidification and how remains unclear. We addressed these questions by re-examining transcriptome and genome data of *Acropora millepora* coral holobionts from volcanic CO$_2$ seeps with end-of-century pH levels. We show that adaptation to ocean acidification is a wholistic process involving the three main compartments of the coral holobiont. We identified 441 coral host candidate adaptive genes involved in calcification, response to acidification, and symbiosis; population genetic differentiation in dinoflagellate photosymbionts; and consistent transcriptional microbiome activity despite microbial community shifts. Coral holobionts from natural analogues to future ocean conditions harbor beneficial genetic variants with far-reaching rapid adaptation potential. In the face of climate change, these populations require immediate conservation strategies as they could become key to coral reef survival.

[1] University of Guam Marine Laboratory, 303 University Drive, 96923 Mangilao, Guam, USA. [2] Departament de Biologia Evolutiva, Ecologia i Ciències Ambientals, Facultat de Biologia, Universitat de Barcelona, Av. Diagonal 643, 08028 Barcelona, Spain. [3] Institut de Recerca de la Biodiversitat (IRBio), Universitat de Barcelona, Barcelona, Spain. ✉email: cleivama@gmail.com

In the last 250 years, atmospheric partial pressure of carbon dioxide ($pCO_2$) increased by 50% due to anthropogenic fossil fuel combustion, representing the fastest $pCO_2$ change on Earth in millions of years[1]. Over this period, oceans absorbed around a third of the total anthropogenic $CO_2$ emissions[2,3], resulting in a pH reduction and a decrease of carbonate saturation in surface waters commonly referred to as ocean acidification[4].

Many calcifying marine organisms such as reef-building corals, whose skeletons provide the scaffolding for the most biodiverse marine habitat (i.e., coral reefs), depend on carbonate to build their calcium carbonate structures[5]. Although ocean warming's devastating effects are already evident in coral reefs throughout the tropics, ocean acidification is predicted to have major consequences for marine calcifying organisms in the following decades by increasing both the dissolution of present calcium carbonate skeletons and the difficulty of depositing new skeleton[6]. Moreover, the combined effects of global ocean acidification and warming appear to increase global reductions in net carbonate production and accretion of most coral reefs[7].

According to these predictions, many reefs will degrade within the following decades unless corals are able to keep pace with the foreseen global environmental changes. Corals' responses are limited to two main processes: poleward range expansions or persistence through adaptation and/or acclimatization[8]. Examples of poleward range expansions of tropical coral species have been reported in southern Japan in response to rising sea temperatures[9], while mechanisms of rapid acclimatization of corals are still being unraveled. Rapid acclimatization usually takes place at the holobiont level and may include photosymbiont shuffling, bacterial community shifts and epigenetic modifications transmitted to the progeny through transgenerational plasticity[10]. Although acclimatization mechanisms will surely play an important role on coral long-term survival, the amount of standing genetic variation and the presence of adaptive alleles in coral metapopulations will be decisive in future coral reef persistence[11,12].

Underwater volcanic $CO_2$ seeps act as natural analogues of a future acidified ocean and provide unique opportunities to study long-term acclimatization and adaptation to ocean acidification[13,14]. Coral populations occurring in these in situ natural laboratories could harbor adaptive alleles conferring them with high resilience and/or resistance to ocean acidification. These adaptive alleles may have been selected by natural selection like in coral colonies living in naturally high-temperature microclimates[15]. For instance, temperate azooxanthellate solitary Mediterranean corals from a $CO_2$ seep system of Italy exhibit high genetic differentiation in genes involved in calcification[16]. However, no study to date has used $CO_2$ seeps to study genomic signals of local adaptation to ocean acidification in tropical reef-building corals.

Here, we used a publicly available transcriptomic dataset (from Kenkel and coauthors[17]) together with recent high-quality reference genomes to (i) unveil genomic targets of natural selection in the coral host, (ii) outline algal symbiont population differentiation and (iii) assess microbiome functionality changes in the reef-building coral *Acropora millepora* from the $CO_2$ seep system of Milne Bay Province in Papua New Guinea (Fig. 1). This $CO_2$ seep system has been relatively well studied, with changes reported in diversity and composition of corals and reef-associated macroinvertebrate communities at Dobu and Upa-Upasina reefs[18,19]. Seep and control sites have been identified at both Dobu and Upa-Upasina reefs, with pH ranging from 7.72 to 7.81 in the seep sites and from 7.98 to 8.01 in the control sites, respectively (see Methods and Fabricius and coauthors[18] for a thorough description of the study site). Kenkel and coauthors[17] compared gene expression profiles of *Acropora millepora* and their algal symbionts from control and seep environments in

Dobu and Upa-Upasina reefs, identifying core transcriptomic responses involved in long-term acclimatization to ocean acidification. Here, we called Single Nucleotide Polymorphisms (SNPs) from Kenkel and coauthors'[17] transcriptomic data using a recently sequenced high-quality reference genome of *A. millepora*[20] to uncover genomic signals of natural selection and local adaptation to high $pCO_2$ conditions. SNPs associated with environmental variables and SNPs showing signals of natural selection provide insight into the particular genomic regions that are under selective pressure, and have commonly been used to inform about the adaptive potential of coral species[21]. We hypothesize that these signals exist in genes involved in calcification and skeleton formation processes, on top of changes in gene expression. Symbiodiniaceae community composition in *A. millepora* does not differ between coral colonies from $CO_2$ seeps and control environments in the seep system of Milne Bay Province, with *Cladocopium goreaui* (previously known as *Symbiodinium* Clade C, type C1) being the dominant species found in all coral colonies[17,21]. However, we hypothesize that intraspecific genetic differences exist between $CO_2$ seep and control *C. goreaui* populations due to environmental or coral host selection of specific *C. goreaui* genotypes. To test this hypothesis, we called SNPs using *C. goreaui* transcriptomic data from Kenkel and coauthors[17] with a pool-seq approach, and performed population genetic analyses. Finally, Morrow and coauthors[22] identified microbial community changes driven by a reduction of symbiotic *Endozoicomonas* in *A. millepora* from $CO_2$ seeps at Upa-Upasina reef. We hypothesize that these microbial community changes also drive changes in microbial transcriptional activity and extracted microbiome metatranscriptome from Kenkel and coauthors'[17] transcriptome data to unveil specific functions that might differ between *A. millepora* microbiomes from $CO_2$ seep and control sites.

## Results

**Candidate adaptive SNPs and population structure of the coral host.** The data downloaded from SRA project PRJNA362652 consisted of a total of 316.8 million raw reads and an average of 5.4 million sequences per individual[17]. A total of 40.7 million clean reads mapped the *Acropora millepora* reference genome and passed post-mapping filters, averaging 690,440 reads per individual, and 79,273 SNPs were obtained after variant calling and variant filtering. From these 79,273 loci, 977 SNPs were associated with pH in the RDA (Fig. 2a), 656 SNPs were detected as under selection by BayPass *in basic mode* (Fig. 2b), 3,235 SNPs were associated with pH in the BayPass run *in covariate mode* (Fig. 2c), and 35 SNPs were detected as under diversifying selection by BayeScan (Fig. 2d). The overlap of these four SNP lists resulted in 625 loci that were identified by at least two of the four analyses performed (Fig. 2e), which were then considered as candidate adaptive SNPs.

Population structure analyses of the candidate SNPs under selection showed genetic differentiation among sites according to their pH value. The DAPC showed an ordination of the samples on the first DAPC axis following a pH gradient (Fig. 3a). Similarly, STRUCTURE results showed that cluster assignment proportions were driven by pH (Fig. 3c).

A dataset of 11,169 independent neutral SNPs was obtained after removing candidate adaptive SNPs and physically linked loci. Analyses of population structure and connectivity of the independent neutral SNP dataset contrastingly showed homogeneity and panmixia among the four sampling sites (Fig. 3b, d). In agreement with these results, global $F_{ST}$ value was low (0.002) and the migration network analysis revealed high levels of connectivity among the four sites (Supplementary Fig. 1).

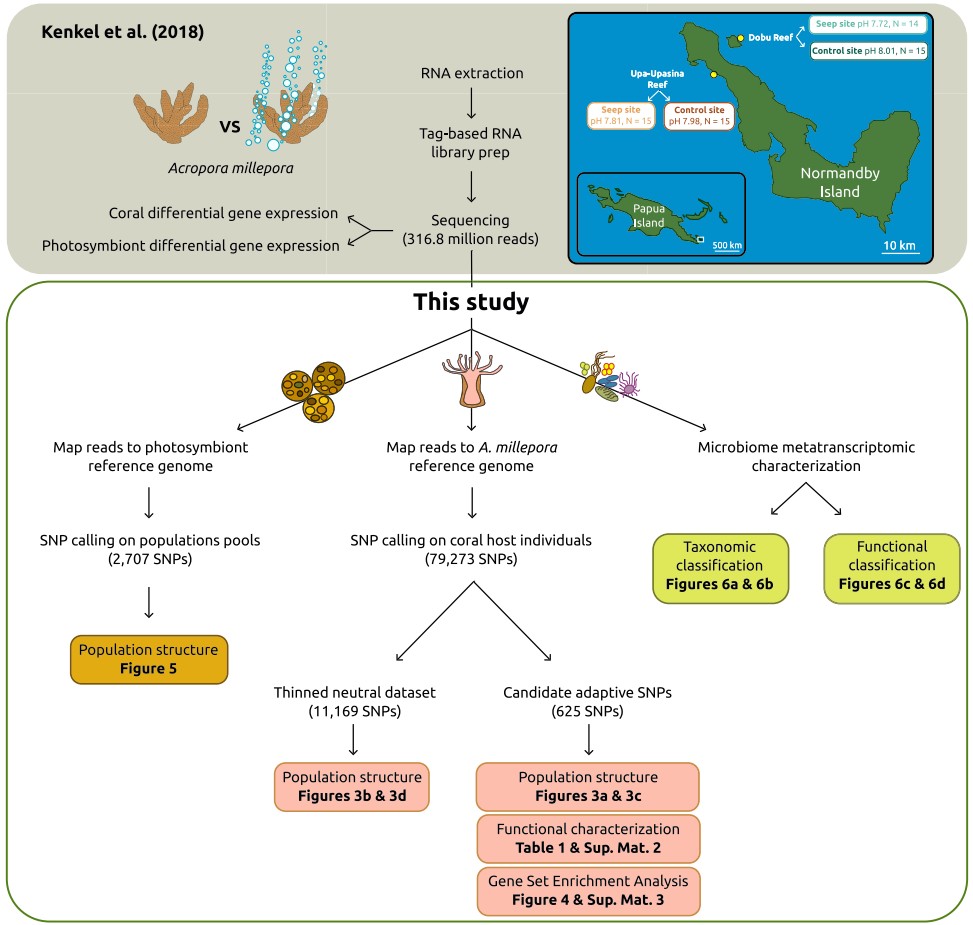

**Fig. 1 Schematic workflow of Kenkel and coauthors[17] and our study with a map of Normandby and Dobu Islands showing site locations, pH value and number of colonies (N) collected.** Inset map shows the location of Normandby and Dobu Islands within the Papua Island region (white square). Sampling sites are colored as follows: teal for Dobu Reef, brown for Upa-Upasina Reef; darker colors for control sites, lighter colors for seep sites. Kenkel and coauthors'[17] schematic workflow is presented at the top of the figure, shaded in gray. The schematic workflow of this study is below with the analyses conducted for the three holobiont compartment detailed: photosymbionts on the left, coral host in the center and microbiome on the right.

**Loci under selection in the coral host.** The combination of *A. millepora* reference genome annotations with blast searches resulted in the annotation of 589 out of the 625 candidate adaptive SNPs, which were located in 441 genes (Supplementary Data 1). The 441 candidate adaptive genes included genes with key functions in the calcification process and coral response to acidification, genes involved in the establishment and maintenance of symbiosis, and genes in common with other coral adaptation studies (Table 1).

The enrichment analysis identified 528 enriched GO terms among the candidate adaptive genes (Supplementary Data 2). The top 20 enriched GO terms for each category included the Biological Processes "regulation of adherens junction organization" and "immunological memory formation process", the Cellular Component "9 + 2 motile cilium", and the Molecular Function "coreceptor activity involved in Wnt signaling pathway" (see Fig. 4 for complete list).

**Cladocopium goreaui population differentiation.** A total of 25.8 million reads mapped the *C. goreaui* reference genome, with an average of 437,714 reads per population pool (i.e., coral colony). A total of 2,707 SNPs were retained after SNP calling and filtering.

Population differentiation analyses of the *C. goreaui* pool-seq SNP dataset showed genetic differentiation between reefs and environments (Fig. 5). The DAPC showed that samples segregated by reef on the first DAPC axis (Fig. 5a, b). Samples from Dobu and Upa-Upasina split by environment on the second (Fig. 5a) and the third (Fig. 5b) DAPC axes, respectively. Pairwise $F_{ST}$ analyses showed that *C. goreaui* population pools tend to present lower genetic differentiation within reef and within environment, and highlighted the presence of a highly differentiated seep-specific genetic cluster in Dobu Seep (Fig. 5c). In agreement with these results, the AMOVA showed that both space and environment explained a small but significant part of the genetic variation ($p$-values < 0.01) (Table 2).

**Microbiome functional characterization.** A total of 702,876 reads were taxonomically classified with Kraken2 using the NCBI taxonomic information, all of them belonging to Bacteria. The non-metric multidimensional scaling (NMDS) plot on the Bray-Curtis dissimilarities among colonies showed no clustering of samples, indicating similar abundances of metatranscriptomic reads from each taxonomic group regardless of reef and environment (Fig. 6a). In fact, the four sites presented similar percentages of metatranscriptomic reads from each phylum (Fig. 6b). Proteobacteria and Firmicutes were the dominant phyla representing ~50% and ~30% of the metatranscriptomic reads at each site, respectively, followed by Actinobacteria, Bacteroidetes, Cyanobacteria and Acidobacteria each accounting for ~3–6% of the metatranscriptomic reads.

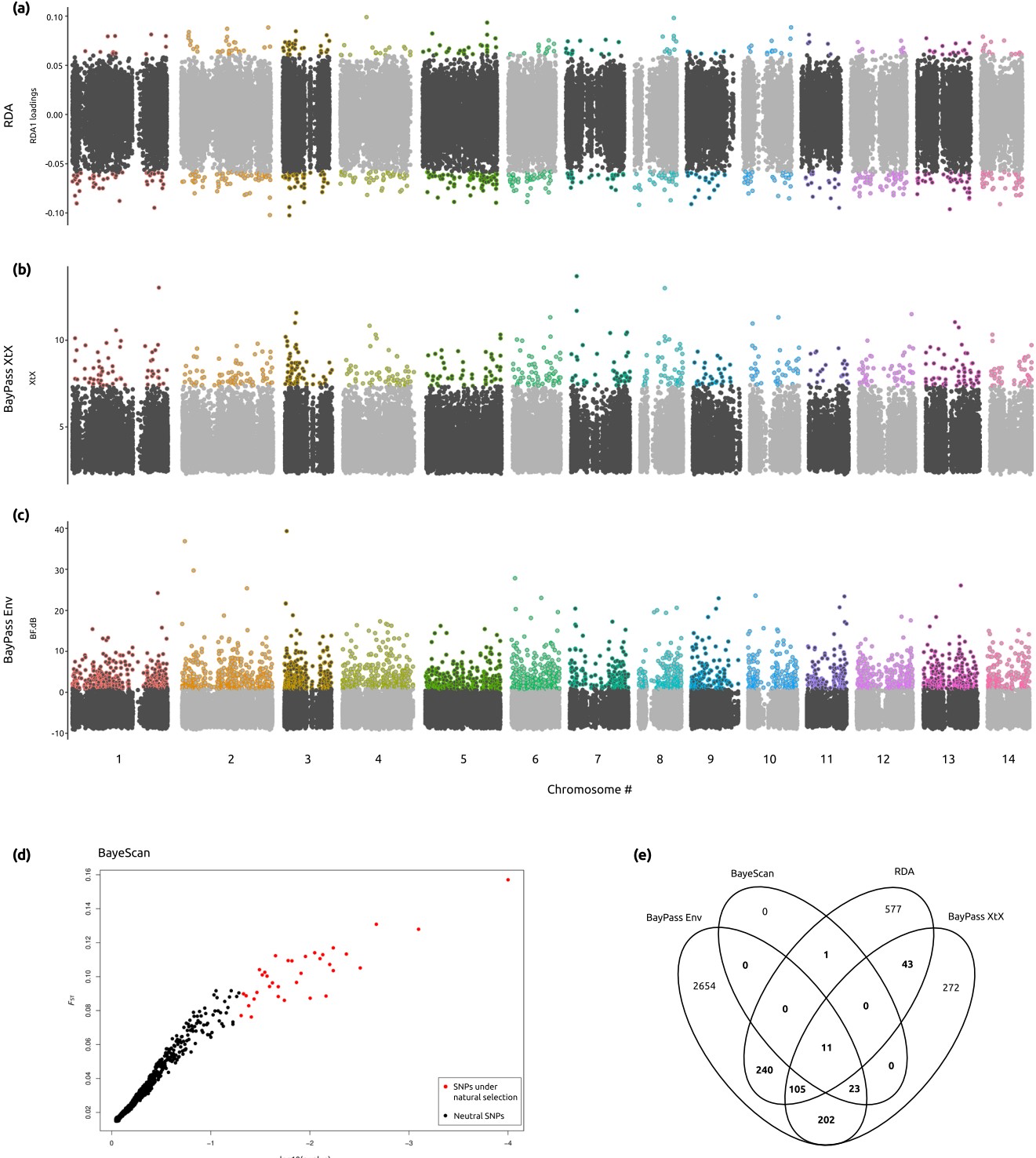

**Fig. 2 Genome scans and candidate adaptive SNPs to low pH environment in the coral host.** For the Manhattan plots (**a**–**c**) dark gray and light gray circles represent SNPs in odd and even chromosomes, respectively, loci under selection or associated with pH are represented by circles with strokes colored by chromosome. **a** Redundancy Analysis (RDA). **b** BayPass run *in basic mode* (BayPass XtX). **c** BayPass run *in covariate mode* (BayPass Env). **d** BayeScan results showing SNPs under diversifying selection in red circles. **e** Venn diagram representing overlapping results among the four analyses. The 625 SNPs identified by at least two of the four analyses are represented in bold.

A total of 318,320 reads were functionally classified with Kraken2, all of them belonging to Bacteria. As for the taxonomic classification, the NMDS plot based on the functional classification showed no clustering of samples, indicating similar abundances of metatranscriptomic reads from each GO term regardless of reef and environment (Fig. 6c). Significant differences in read counts for each GO term were tested with Welch t-tests. After applying a Bonferroni correction, no GO term presented significantly different read counts between seep and control colonies.

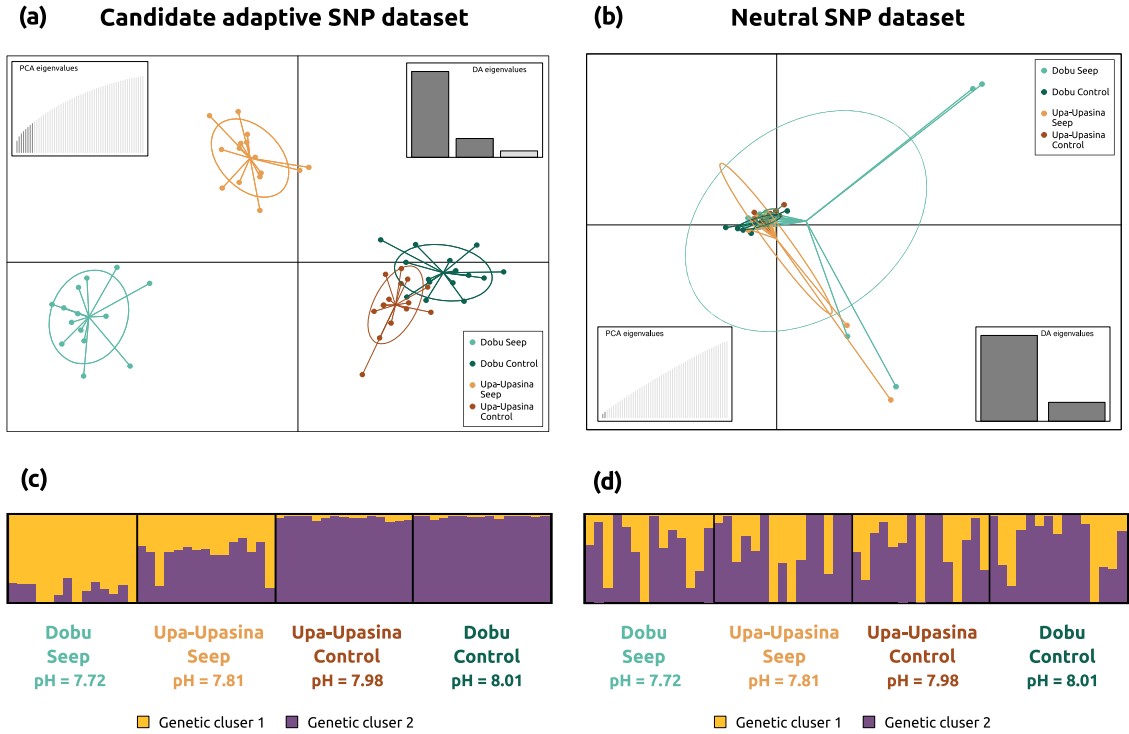

**Fig. 3 Population genomic structure of the coral host. a** DAPC results using the dataset of 625 candidate adaptive SNPs. **b** DAPC results using the dataset of 11,169 independent neutral SNPs. Both DAPC plots are colored following Fig. 1. **c** STRUCTURE results with $K = 2$ for the candidate adaptive SNP dataset (see delta $K$ plot in Supplementary Fig. 2b). **d** STRUCTURE results with $K = 2$ for the independent neutral SNP dataset (see delta $K$ plot in Supplementary Fig. 2a).

## Discussion

Our functional genomic analyses revealed that natural selection in a subset of coral host genes with key functions in calcification and skeleton formation appeared to be driving adaptation to ocean acidification in *Acropora millepora* in the $CO_2$ seep system of Milne Bay Province. For instance, carbonic anhydrases play major roles in the physiology of coral calcification[23,24] by catalyzing the interconversion of $CO_2$ to bicarbonate ions and protons. Endothelin-converting enzyme, and CUB and peptidase-domain containing protein are part of the coral skeleton proteome[25,26] and are involved in membrane processing, extracellular matrix/transmembrane and protein modification processes, as well as vesicle/secretion and metal binding[26]. Alongside these three enzymes, a relevant amount of genes encoding for coral skeletal proteins such as α-collagen, Protocadherin, Coadhesin, and Complement C3, also presented signals of adaptation to seep environments (Table 1). Additionally, signals of selection were detected in genes involved in the response to acidification (Table 1). These genes included *Carbonic anhydrase 2*, *Fibropellin-1*, *Bone morphogenetic protein 7* and *V-type proton ATPase subunit S1*, which were differentially expressed in corals exposed to elevated $pCO_2$[27–29].

Three of the main representative enriched GO terms, "regulation of adherens junction organization", "9 + 2 motile cilium" and "coreceptor activity involved in Wnt signaling pathway" are intimately related to coral calcification. Intercellular junctions determine the ectodermal tissue permeability, regulating biomineralization by controlling the diffusion of molecules that can be transferred to the Extracellular Calcifying Medium (ECM)[30,31]. Enrichment of the GO term "9 + 2 motile cilium" suggests an important role in coral calcification of the cilia present on the ECM membrane of the coral calcifying cells[32] Finally, the Wnt signaling pathway is a classical developmental signaling pathway that, in corals, controls development, growth, and calcification[33–35]. Our results demonstrate that despite the lack of population structure in the neutral SNP dataset, natural selection is occurring in core molecular pathways that control biomineralization, such as enzymes, skeletal proteins and ion channels. These results show that local adaptation is independent of migration under spatially varying selection when selective pressure is strong enough to counteract the effects of migration[15,36–39]. GO terms related to the immune system such as the BP "immunological memory formation process" and "response to other organism", were also enriched in the list of candidate adaptive genes. Kenkel and coauthors[17] suggested a potential impact of long-term acidification on the innate immune response of *A. millepora* at the $CO_2$ seeps. Particularly, they found that the *TNF receptor-associated factor* (ig7766), which is involved in the innate immune response and the coral stress response[40], was a top discriminatory gene for seep environment[17]. Interestingly, signals of selection and associations with seep environments at the *TNF receptor-associated factor* were found in the present study, confirming that changes in its expression are part of an adaptation mechanism to long-term elevated $pCO_2$. Signals of environmental-driven adaptation were also found in eight additional differentially regulated loci from Kenkel and coauthors[17], suggesting *cis*-regulation of gene expression in key components of the coral response to acidification, including calcification, transcription processes, and the immune system (Table 1).

Previous studies scanning for genomic signals of local adaptation and genotype-environment associations on reef-building corals were mostly focused on disentangling their adaptive responses to elevated temperature[15,20,41,42]. We found that some of the candidate adaptive genes here were shared with genes locally adapted in these studies, suggesting common targets of natural selection across environmental stressors. For instance, we found seven genes in common between our candidate adaptive genes and those conferring heat resistance in *Acropora*

**Table 1 Relevant genes with signals of adaptation to acidification detected in *Acropora millepora*, grouped by function.**

| | References | Coral species studied |
|---|---|---|
| **Calcification and skeleton formation** | | |
| Carbonic anhydrase 2 | Goreau[23]; Bertucci et al.[24]; Ramos-Silva et al.[25]; Peled et al.[26] | Multiple species |
| Endothelin-converting enzyme | Peled et al.[26] | *Stylophora pistillata* |
| CUB and peptidase-domain containing protein | Peled et al.[26] | *Stylophora pistillata* |
| Bone morphogenetic protein 7 | Zoccola et al. (2008)[108] | Multiple species |
| a-collagen | Ramos-Silva et al.[25] | *Acropora millepora* |
| Protocadherin | Drake et al.[109]; Ramos-Silva et al.[25] | *Stylophora pistillata; Acropora millepora* |
| Coadhesin | Ramos-Silva et al.[25] | *Acropora millepora* |
| Complement C3 | Peled et al.[26] | *Stylophora pistillata* |
| Polycystic kidney disease 1-related protein | Ramos-Silva et al.[25] | *Acropora millepora* |
| CUB and peptidase-domain containing protein 1 | Ramos-Silva et al.[25]; Peled et al.[26] | *Stylophora pistillata* |
| Mammalian ependymin-related protein 1 | Peled et al.[26] | *Stylophora pistillata* |
| ZP domain-containing protein | Peled et al.[26] | *Stylophora pistillata* |
| Endothelin-converting enzyme homolog | Peled et al.[26] | *Stylophora pistillata* |
| Uncharacterized skeletal organic matrix protein 6 | Ramos-Silva et al.[25] | *Acropora millepora* |
| **Coral response to acidification** | | |
| Carbonic anhydrase 2 | Moya et al.[27]; Vidal-Dupiol et al.[29] | Acropora millepora; Pocillopora damicornis |
| Fibropellin-1 | Moya et al.[27] | *Acropora millepora* |
| Tenascin-X | Moya et al.[27] | *Acropora millepora* |
| Leucine-rich repeats and immunoglobulin-like domains protein 1 | Moya et al.[27] | *Acropora millepora* |
| Low-density lipoprotein receptor-related protein 6 | Moya et al.[27] | *Acropora millepora* |
| Transient receptor potential cation channel subfamily M member-like 2 | Moya et al.[27] | *Acropora millepora* |
| Bone morphogenetic protein 7 | Vidal-Dupiol et al.[29] | *Pocillopora damicornis* |
| V-type proton ATPase subunit S1 | Kaniewska et al.[28] | *Acropora millepora* |
| **Establishment and maintenance of symbiosis** | | |
| Calumelin | Mohamed et al.[54] | *Acropora digitifera* |
| Tetratricopeptide repeat protein 28 | Mohamed et al.[54] | *Acropora digitifera* |
| NADH dehydrogenase [ubiquinone] 1 beta subcomplex subunit 9 | Mohamed et al.[54] | *Acropora digitifera* |
| 60S ribosomal protein L44 | Mohamed et al.[54] | *Acropora digitifera* |
| Transcription factor HES-1 | Mohamed et al.[54] | *Acropora digitifera* |
| 60S ribosomal protein L27 | Mohamed et al.[54] | *Acropora digitifera* |
| Translocon-associated protein subunit gamma | Mohamed et al.[54] | *Acropora digitifera* |
| Reversion-inducing cysteine-rich protein with Kazal motifs | Mohamed et al.[54] | *Acropora digitifera* |
| Heparanase | Yoshioka et al. (2020)[55] | *Acropora tenuis* |
| Golgi-associated plant pathogenesis-related protein 1 | Yoshioka et al. (2020)[55] | *Acropora tenuis* |
| Carbonic anhydrase 2 | Yoshioka et al. (2020)[55] | *Acropora tenuis* |
| DBH-like monooxygenase protein 1 | Yoshioka et al. (2020)[55] | *Acropora tenuis* |
| B-cell lymphoma 3 protein | Yoshioka et al. (2020)[55] | *Acropora tenuis* |
| Acylamino-acid-releasing enzyme | Yoshioka et al. (2020)[55] | *Acropora tenuis* |
| Low-density lipoprotein receptor-related protein 6 | Yoshioka et al. (2020)[55] | *Acropora tenuis* |
| Tachylectin-2 | Kuniya et al.[56] | *Acropora tenuis* |
| Adehyde dehydrogenase | Oakley et al.[57] | *Aiptasia sp.* |
| MAM and LDL-receptor class A domain-containing protein 1 | Oakley et al.[57] | *Aiptasia sp.* |
| **Genes with signals of adaptation that were also differentially expressed between seep and control sites in Kenkel et al.[17]** | | |
| Catenin alpha-2 | Kenkel et al.[17] | *Acropora millepora* |
| Carbonic anhydrase 2 | Kenkel et al.[17] | *Acropora millepora* |
| 60S ribosomal protein L7 | Kenkel et al.[17] | *Acropora millepora* |
| 60S ribosomal protein L12 | Kenkel et al.[17] | *Acropora millepora* |
| TNF receptor-associated factor 6 | Kenkel et al.[17] | *Acropora millepora* |
| Myosin-2 heavy chain | Kenkel et al.[17] | *Acropora millepora* |
| DnaJ homolog subfamily B member 6 | Kenkel et al.[17] | *Acropora millepora* |
| DnaJ homolog subfamily C member 10 | Kenkel et al.[17] | *Acropora millepora* |
| DnaJ homolog subfamily C member 21 | Kenkel et al.[17] | *Acropora millepora* |
| **Genes in common with other adaptation studies in corals** | | |
| Kinesin-like protein KIF16B | Bay & Palumbi[15] | *Acropora hyacinthus* |
| Myosin heavy chain | Bay & Palumbi[15] | *Acropora hyacinthus* |
| LIM domain and actin-binding protein 1 | Bay & Palumbi[15] | *Acropora hyacinthus* |
| Transcription factor HES-1 | Bay & Palumbi[15] | *Acropora hyacinthus* |
| Dystonin | Bay & Palumbi[15] | *Acropora hyacinthus* |
| Calumenin | Bay & Palumbi[15] | *Acropora hyacinthus* |
| Golgin subfamily B member 1-like | Bay & Palumbi[15] | *Acropora hyacinthus* |

**Table 1 (continued)**

|  | References | Coral species studied |
| --- | --- | --- |
| Synaptotagmins | Smith et al.[42] | *Platygyra daedalea* |
| 26S proteasome non-ATPase regulatory subunit | Rose et al.[41] | *Acropora hyacinthus* |
| Ribosomal proteins | Bay & Palumbi[15]; Rose et al.[41] | *Acropora hyacinthus* |

The associated references and other coral species the genes were found in are listed.

*hyacinthus*[15] (Table 1). Additionally, our analyses revealed signals of adaptation in two Synaptotagmins, which also showed significant genetic differentiation between *Platygyra daedalea* colonies from the warm Persian/Arabian Gulf and the relatively cooler Gulf of Oman[42]. Synaptotagmins are membrane-trafficking calcium sensor proteins[43] involved in maintaining calcium homeostasis during the coral stress response[40]. As Synaptotagmins' expression varies with different types of cellular stress[44,45], it is not surprising that they would also be the target of natural selection across different environmental stressors such as heat or elevated $pCO_2$. Similarly, strong signals of differentiation were detected in the 26 S proteasome non-ATPase regulatory subunit in both our *A. millepora* from $CO_2$ seep environments and in *A. hyacinthus* with different bleaching susceptibility[41], albeit on different subunits (26 S proteasome non-ATPase regulatory subunit 7 and 11, respectively). Since the 26 S proteasome degrades damaged ubiquitin-conjugated proteins after any type of cellular stress[46], different stressors may induce a similar selective pressure on the regulation of this protease machinery. Finally, many ribosomal proteins seem to be frequently targeted by natural selection under different long-term environmental stressors[15,41]. Signals of adaptation in ribosomal proteins may be due to the role of translational reprogramming during the stress response[47,48] or to their extra-ribosomal functions as p53 regulators during cellular stress[49].

Unlike whole transcriptome sequencing, tag-based RNAseq only produces short fragments of the transcriptome, complementary to the 3'-end of the transcripts[50]. Hence, our dataset could be missing other primordial gene regions bearing signals of adaptation that might have been detected with whole transcriptome or whole genome sequencing approaches. However, high linkage and low probability of recombination events within a single gene ensure that signals of selection found in tag-based RNAseq transcripts are representative of selection in that whole gene. In fact, because of the high chance of linkage within neighboring genomic regions, it is common practice in whole genome selection scans to produce thinned SNP datasets filtering physically linked SNPs within a few kbp, resulting in a similar exclusion of putative adaptive SNPs[51,52].

Besides revealing signals of selection in the coral host genome, our results also showed how other compartments of the coral holobiont (Symbiodiniaceae photosymbionts and the bacterial microbiome) respond to long-term elevated $pCO_2$. In fact, the pool-seq approach implemented in this study yielded more informative results than other commonly used methods (i.e., ITS2 DGGE profiling and RFLP[17,21]) about the effects of ocean acidification on algal photosymbionts. Here, we detected the presence of a highly differentiated seep-specific genetic cluster of *C. goreaui* in Dobu Seep site, and a combined effect of both space and environment in structuring *C. goreaui* populations, suggesting that symbiont shuffling may well be a long-term strategy for adaptation to acidification in *A. millepora*[53].

Consistent with these results, we found signals of selection in coral host genes involved in the establishment and maintenance of symbiosis. For instance, 15 genes under selection known to be differentially expressed between symbiotic and apo-symbiotic

coral larvae were detected in the $CO_2$ seep colonies[54,55] (Table 1). Additionally, 3 mutations in the gene coding for the Tachylectin-2 protein, a pattern recognition receptor of the innate immune response involved in photosymbiont acquisition[56], were under selection; one of them causing an amino acid change (Ser264Gly). Finally, two additional genes, known to be differentially expressed between symbiotic and apo-symbiotic *Aiptasia* anemones, were under selection in the $CO_2$ seep corals[57] (Table 1). Accordingly, and as we mentioned above, GO terms related to the immune system were enriched among the candidate adaptive genes, which plays major roles in the establishment of the cnidarian-dinoflagellate photosymbiosis[58,59]. These results suggest adaptation in the regulation of photosymbiont communities in response to long-term acidification, which is considered a coral adaptation mechanism to environmental change[10,51,60].

The characterization of the *A. millepora* microbiome revealed highly similar metatranscriptomes in coral colonies regardless of reef and environment, both taxonomically and functionally. These results indicate that despite significant microbial community changes between seep and control colonies[22], the *A. millepora* microbial transcriptional activity is both taxonomically and functionally constant between the two environments. This could mean for instance, that the key functional activity of various taxa, such as the reduced *Endozoicomonas* at seep sites[22], is maintained either by an extra transcriptional effort from the remaining *Endozoicomonas* or by other taxa taking over their functional activity. Significant coral microbiomes metatranscriptomic differences are usually found between healthy and heavily diseased coral colonies[61,62]. The fact that this pattern is not found between seep and control site *A. millepora* microbiomes most likely indicates that we are in the presence of adapted and acclimatized healthy holobionts at both sites. This interpretation is further supported by the minimal gene expression changes found between seep and control sites *A. millepora* coral hosts[17] and the absence of significant differences in skeletal densities between *A. millepora* colonies from Upa-Upasina control and seep sites[63], a 'supertrait' and an easy-to-measure indicator of coral health[64,65].

Overall, our analyses clearly indicate that there are three distinct and synergistic adaptation mechanisms in three different compartments of the *Acropora millepora* holobiont in the $CO_2$ seep system of Milne Bay Province. We found signals of selection in multiple coral host genes involved in calcification and symbiosis, structuring of *Cladocopium goreaui* genetic clusters and microbiome metatranscriptomic responses to long-term acidification. Remarkably, this study shows that *A. millepora* colonies living in $CO_2$ seeps harbor genomic adaptations to the $pCO_2$ levels foreseen by the end of the current century. These adapted natural populations may become key for the persistence of coral reefs, acting as a source of pre-adapted alleles that may facilitate rapid adaptation to climate change on a global scale, particularly in broadcast-spawning species such as *A. millepora*. Regardless, mitigation of anthropogenic $CO_2$ emissions together with active conservation and restoration actions are essential for coral reef survival[66], as adaptive responses alone will most likely be insufficient under the current rates of environmental change[67] and the multi-factorial nature of climate change.

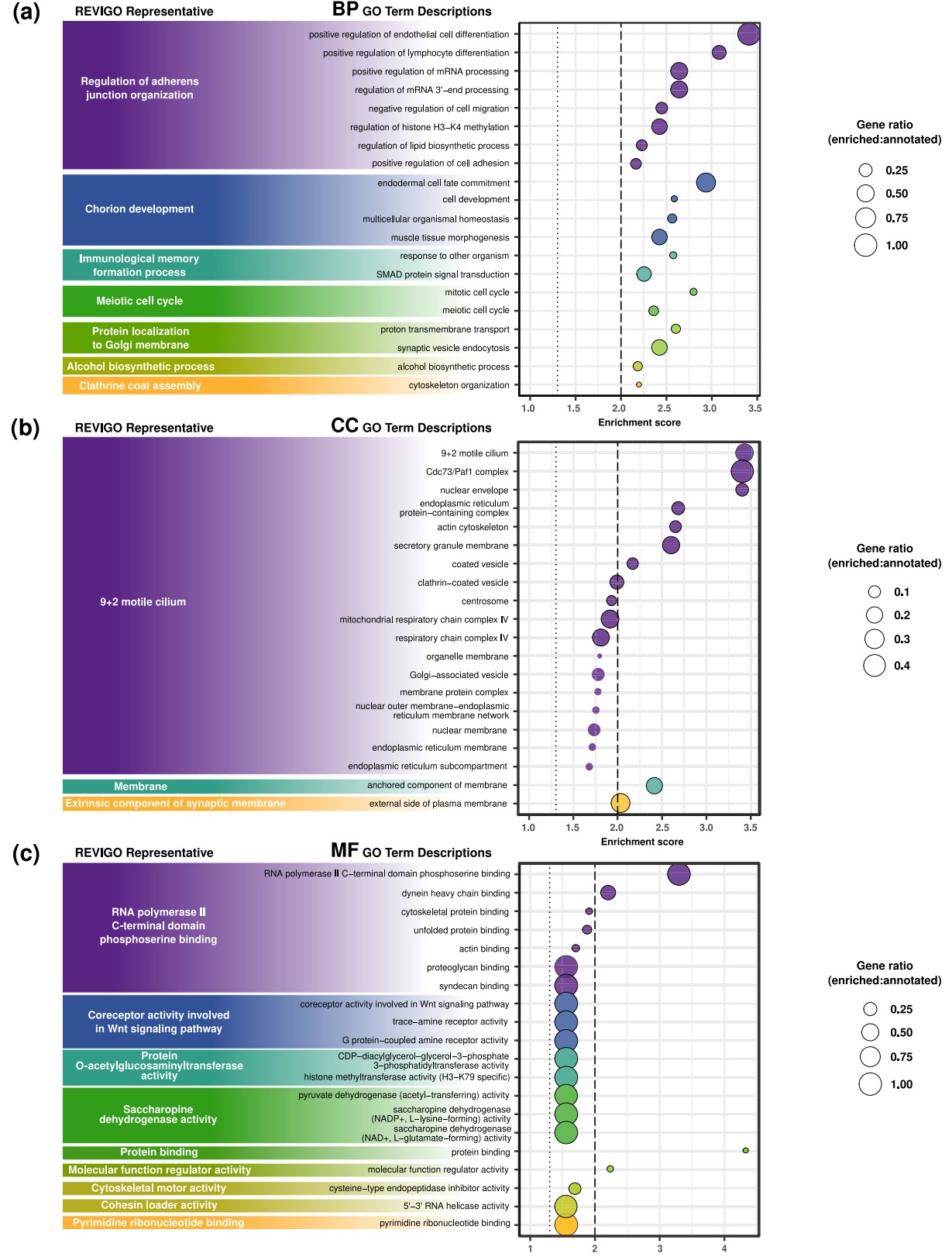

**Fig. 4 Top 20 enriched GO terms for each category, grouped by REVIGO representative (see REVIGO treemaps in Supplementary Fig. 3). a** Biological processes, **b** cellular components, and **c** molecular functions. Enrichment scores (-log10($p$-value)) are plotted for each GO term, with circle size representing the ratio between the number of candidate adaptive genes with a particular GO annotation and the number of genes in the *A. millepora* proteome with that GO annotation. Dotted lines represent $p$-value = 0.05, dashed lines represent $p$-value = 0.01. Numerical source is provided in Supplementary Data 3.

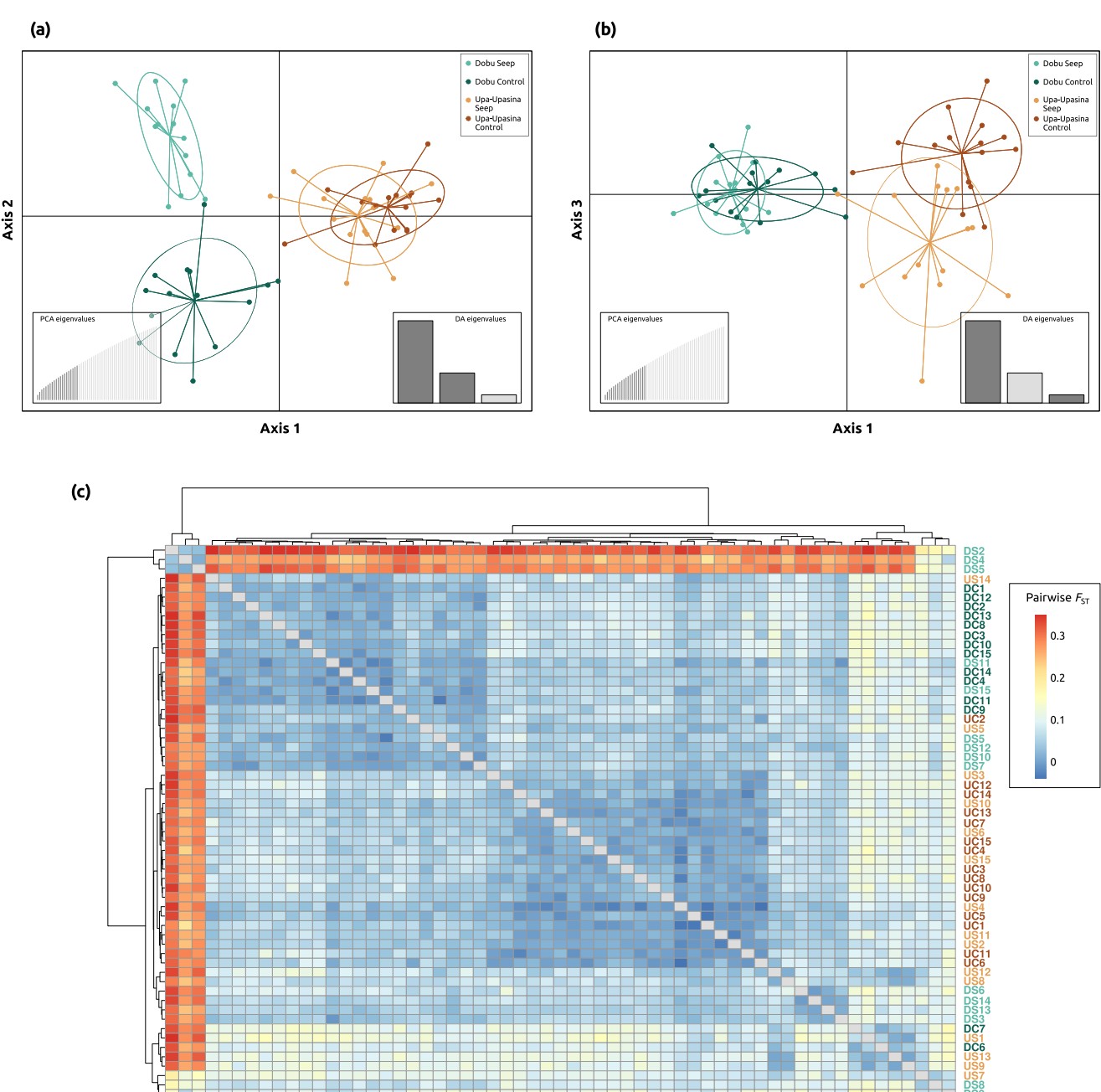

**Fig. 5 Population genetic structure among *Cladocopium goreaui* population pools (i.e., coral colonies).** DAPC results are shown on the first and second (**a**), and the first and third axes (**b**). **c** Heatmap plot of the pairwise $F_{ST}$ matrix among *C. goreaui* population pools. Warm colors represent higher pairwise $F_{ST}$ values, indicating high differentiation between population pools. Cold colors represent lower pairwise $F_{ST}$ values, indicating low differentiation between population pools. Population pools are colored following Fig. 1 and coded in (**c**) as follows: Dobu Seep, DS; Dobu Control, DC; Upa-Upasina Seep, US; Upa-Upasina Control, UC.

## Methods

**Study area, data processing and variant detection in the coral host genome.** Milne Bay Province in Papua New Guinea hosts a well-studied shallow-water seep system of ~99% $CO_2$ gas[17–19,21,22] (see Fabricius and coauthors[18] for a thorough description of the study site). Kenkel and coauthors[17] (2018) sequenced tag-based RNAseq libraries for a total of 59 *Acropora millepora* colonies originating from two reefs in this system, Upa-Upasina and Dobu reefs. At each site, they sampled coral fragments from both a $CO_2$ seep site and an adjacent control site <2.5 km apart: 14 colonies from Dobu $CO_2$ seep site (pH = 7.72, 998 µatm $pCO_2$), 15 from Dobu control environment (pH = 8.01, 368 µatm $pCO_2$), 15 from Upa-Upasina $CO_2$ seep site (pH = 7.81, 624 µatm $pCO_2$), and 15 from Upa-Upasina control environment (pH = 7.98, 346 µatm $pCO_2$)[17] (Fig. 1). Control and seep sites had similar seawater temperature, salinity and geomorphology[18].

Raw reads were downloaded from the NCBI SRA BioProject PRJNA362652[68] and Single Nucleotide Polymorphisms (SNPs) from tag-based RNAseq data were called following GATK best practices and Rogier and coauthors[69]. First, raw reads were adapter and quality trimmed using specific perl scripts for tag-based RNAseq with default parameters (available at https://github.com/z0on/tag-based_RNAseq), followed by a quality check using fastqc 0.11.9[70]. Clean reads were then mapped against the *A. millepora* high-quality reference genome[20,71] using bwa-mem 0.7.17[72], and sorted with samtools 1.16[73]. MarkDuplicates 2.18.25 in Picard tools (http://broadinstitute.github. io/picard/) was used to mitigate biases introduced during library preparation and minimize gene expression variations. Finally, SplitNCigarReads in GATK 4.2.4.1[74] was applied as the last post-processing step as recommended by GATK best practices.

Prior to calling variants with FreeBayes 1.1[75], sample names were added to each original individual BAM file and merged into one single stream with bamaddrg

**Table 2 AMOVA results for the photosymbiont *Cladocopium goreaui* population differentiation following two models: (a) hierarchical AMOVA for the spatial model, (b) one-factor AMOVA for the environmental model.**

|  | Sum Sq | Mean Sq | *df* | Percentage of variation | *p-value* |
|---|---|---|---|---|---|
| **(a) Spatial model: genetic variance ~ reef/site** | | | | | |
| Between Reefs | 2854.09 | 2854.09 | 1 | 2.05 | **0.001** |
| Between Sites within Reefs | 3277.45 | 1638.72 | 2 | 1.68 | **0.001** |
| Between Colonies within Sites | 62656.25 | 1139.20 | 55 | 16.92 | **0.001** |
| Within Colonies | 47121.75 | 798.67 | 59 | 79.35 | **0.001** |
| Total | 115909.54 | 990.68 | 117 | 100 | |
| **(b) Environmental model: genetic variance ~ environment** | | | | | |
| Environment | 1717.81 | 1717.81 | 1 | 0.92 | **0** |
| Between Colonies within Environment | 67069.98 | 1176.67 | 57 | 18.96 | **0** |
| Within Colonies | 47121.75 | 798.67 | 59 | 80.12 | **0** |
| Total | 115909.54 | 990.68 | 117 | 100 | |

Significant *p*-values (<0.05) are highlighted in bold.

## Taxonomic classification

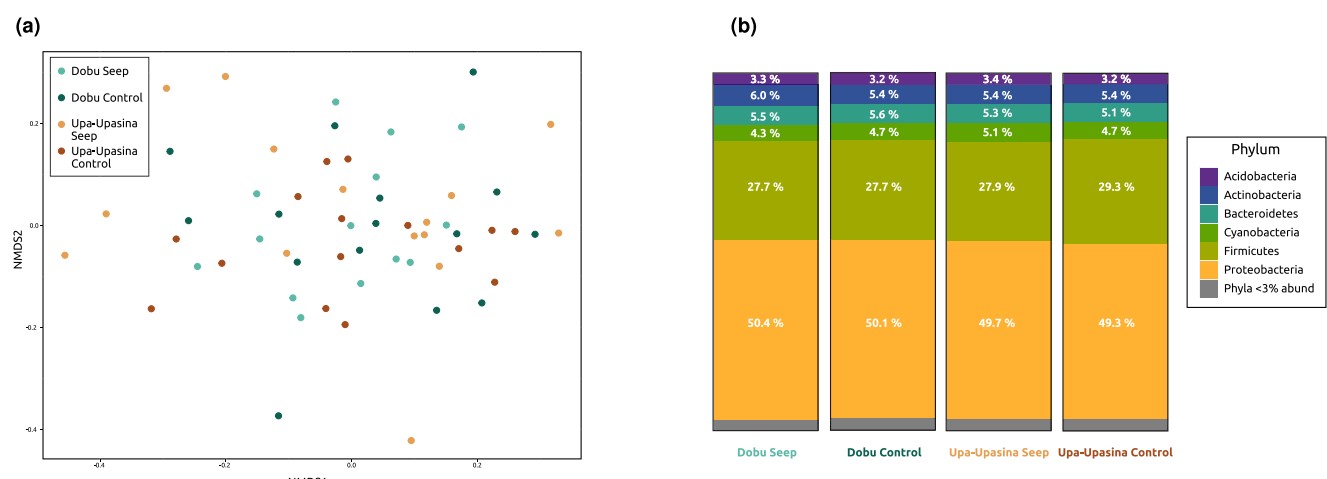

## Functional classification

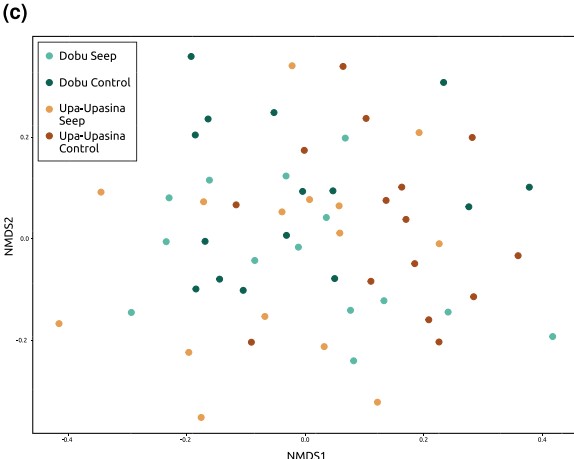

**Fig. 6 Microbiome metatranscriptomic characterization results for the taxonomic and functional classifications. a** NMDS plot on the Bray-Curtis dissimilarities among colonies using the taxonomic classification of metatranscriptomic reads. **b** Percentages of microbial metatranscriptomic reads belonging to each Phylum at each sampling site. **c** NMDS plot on the Bray-Curtis dissimilarities among colonies using the functional classification of metatranscriptomic reads. Coral colonies are colored following Fig. 1. Numerical source is provided in Supplementary Data 4.

(https://github.com/ekg/bamaddrg). Only variants belonging to reads with mapping quality >20 were retained, using the FreeBayes flag -m 20. VCFtools 0.1.16[76] was used to filter and retain biallelic SNPs (--remove-indels --min-alleles 2 --max-alleles 2) with a minimum quality score of 20 (--minQ 20) and present in at least 70% of the individuals (--max-missing 0.7). A minimum allele frequency filter of 0.05 (--maf 0.05) was set to avoid calling singleton SNPs. Our final *A. millepora* coral host dataset consisted of 79,273 SNPs.

**Identification and annotation of loci under selection and genotype-environmental association analyses in the coral host**. Four approaches were implemented to identify potential adaptive SNPs associated with pH: BayeScan 2.1[77], BayPass 2.3[78] *in basic mode*, BayPass *in covariate mode*, and a Redundancy Analysis (RDA) using the *rda* function in the vegan R package[79]. Due to the negligible population structure found among sites (see Results and Fig. 3) and in an effort to increase the number of individuals per condition as well as the model accuracy, individuals from both control sites and from both seep sites were pooled for the BayeScan and BayPass *in basic mode* analyses.

BayeScan was run with default parameters, using the -snp option, and a false discovery rate of 0.05 (FDR = 0.05). To control for potential population structure in the BayPass *in basic mode* analysis, a Linkage Disequilibrium (LD) pruned dataset was first obtained using plink 1.9[80] and plink 2.0[80]. A first run was performed with the LD pruned dataset, after which the resulting omega matrix was used for the controlled run with the whole SNP dataset. To establish the threshold of the XtX statistic to consider a SNP as under selection for the BayPass *in basic mode*, a neutral SNP distribution was simulated with the *simulate.baypass* R function included in the BayPass package. Then, BayPass was run on the simulated data and, finally, SNPs under selection with an FDR = 0.05 were obtained by selecting the 95% quantile of the simulated XtX distribution.

To identify genotypes associated with pH changes, BayPass *in covariate mode* and an RDA were performed, using the pH value of each site as the covariate environmental data for both analyses. For the RDA, missing genotypes were first imputed with the most common genotype at each site, and the *rda* function was run on the imputed dataset. SNP loadings were extracted for the RDA axis 1, and the outlier threshold was established as 2.5 times the standard deviation, equaling a $p = 0.01$.

To avoid false positives, a SNP was considered adaptive if it appeared at least in two of the four analyses performed. This resulted in a total of 625 candidate adaptive SNPs, which were annotated using the reference genome annotations and SnpEff[81]. As tag-based RNAseq produces fragments of the transcriptome near the 3'-end of the transcripts[50], many reads were expected to map 3'-UTR regions, which are generally poorly annotated in reference genomes. Indeed, 16.7% of the SNPs were annotated as intergenic SNPs by SnpEff using *A. millepora* genome annotations from Fuller and coauthors[20]. To improve annotations of these transcripts, BLASTN and BLASTX searches[82] against the *nr* NCBI database were performed using default parameters. Although the exact mechanisms by which these genes operate might still be unknown, differentially expressed genes in transcriptomic studies are functionally involved in the response to the environment they were exposed to. Thus the function of the annotated candidate adaptive SNPs was investigated and when possible, determined using numerous previously published transcriptomic and functional coral studies.

A Gene Set Enrichment Analysis (GSEA) was performed to test whether there were over-represented Gene Ontology (GO) terms in the candidate adaptive SNP annotations compared to the whole genome. The proteome fasta file of the *A. millepora* reference genome was annotated using the PANNZER2 web server[83]. Then, a GSEA was performed with the *GOEnrich_pannzer2* function in the PlantNGSTools R package[84], using the GO annotations obtained with PANNZER2 and the list of candidate adaptive genes. Following the original description of the GSEA method[85], only those gene sets with a false discovery rate <0.25 were selected ("useFDR = T, cut = 0.25") in order to control for false positives. The REVIGO web server[86] was used to cluster, remove redundant, and find representative enriched GO terms for each GO category (Biological Process, BP; Cellular Component, CC; Molecular Function, MF). Although some GO terms should be interpreted with caution in non-model organisms, the species-neutrality design of the GO database specifically allows the transfer of functional annotations from model genes to their non-models orthologues[87].

**Coral host population genetic analyses**. To verify if corals from the two sampling sites were genetically distinct or homogeneous populations, population genetic analyses were performed. A neutral SNP dataset of 78,648 SNPs was obtained by excluding the 625 candidate adaptive SNPs using the VCFtools function --exclude-positions. Subsequently, to avoid the inclusion of physically linked loci, the neutral SNP dataset was thinned to ensure physical distance of at least 10 kbp (following *e.g.*, Bennett and coauthors[88]; Mattingsdal and coauthors[52]; Zou and coauthors[89]) using the function --thin in VCFtools, resulting in a dataset of 11,169 independent neutral SNPs.

Population structure in both the independent neutral and the candidate adaptive SNP datasets was assessed using the Bayesian clustering approach in STRUCTURE 2.3[90] and the discriminant analyses of principal components (DAPC) implemented in the adegenet R package[91]. STRUCTURE was run for 200,000 MCMC iterations with a burn-in of 50,000 iterations, using the admixture

model and setting the putative number of clusters (K) from 1 to 10 with 10 replicates for each run. STRUCTURE HARVESTER[92] was used to find the most likely number of genetic clusters using the Evanno Delta K method[93]. Subsequently, CLUMPAK[94] was used to average individual's membership across replicates and graphically display STRUCTURE results. The DAPC analysis was performed using the sampling sites as a prior and choosing the optimal number of retained principal component axes using the *xvalDapc* function in the adegenet R package. Two and eight PC axes and two and three DA axes were retained for the neutral and the adaptive SNP datasets, respectively.

Additionally, for the independent neutral SNP dataset, a migration network was inferred using Nei's $G_{ST}$ method of the *divMigrate* function in the diveRsity R package[95], and the global $F_{ST}$ value was calculated using the *basic.stats* function in the hierfstat R package[96].

**Cladocopium goreaui endosymbiont population genetic analyses using a pool-seq approach**. To test for photosymbiont genetic differentiation between sites, a pool-seq approach was implemented. In this approach, every coral colony corresponded to one population pool of algal endosymbionts. Both Noonan and coauthors[21] and Kenkel and coauthors[17] demonstrated that the Symbiodiniaceae type did not differ between control and $CO_2$ seep environments, with *Cladocopium goreaui* (previously known as *Symbiodinium* Clade C, type C1) being the dominant species in both environments. Hence, clean reads from each colony were mapped against the *Cladocopium goreaui* reference genome obtained from http://symbs.reefgenomics.org[97] using bwa-mem2 v. 2.2.1[98]. Positions with a mapping quality <20 were filtered out with samtools *view* and files were sorted by reference position using samtools *sort*.

SNP calling of *C. goreaui* endosymbionts was performed using FreeBayes with the --pooled-coninuous flag to account for pooled populations of unknown number of individuals. VCFtools was used to filter and retain SNPs (--remove-indels) with a minimum quality score of 20 (--minQ 20), a minimum allele frequency of 0.05 (--maf 0.05) and present in at least 70% of the individuals (--max-missing 0.7). Our final *C. goreaui* photosymbiont dataset consisted of 2,707 SNPs. Population structure was assessed with a DAPC analysis as indicated above for the coral host, retaining 20 PC and three DA axes as optimized by *xvalDapc*. The vcf file was then transformed into a pooldata object using the *vcf2pooldata* function in the poolfstat 2.1.1 R package[99]. A pairwise $F_{ST}$ table was calculated using the *compute.pairwiseFST* function in the poolfstat R package and the results were visualized with the *pheatmap* function in the pheatmap 1.0.12 R package[100]. Additionally, two analyses of molecular variance (AMOVA) were conducted using the *amova* function in the poppr 2.9.3 R package[101] to test the partition of genetic variance in both a spatial and an environmental model, with *p*-values based on 1000 permutations. In the spatial model, the genetic variance was predicted by sampling site (Dobu seep, Dobu control, Upa-Upasina seep, Upa-Upasina control) nested within reef system (Dobu, Upa-Upasina). In the environmental model, the genetic variance was solely predicted by the environment of the population pools (i.e., controls vs $CO_2$ seeps) to test whether the environment has a significant effect on the endosymbiont genetic variance.

**Microbiome taxonomic and functional characterization using a metatranscriptomic approach**. Microbiome transcriptomic reads were taxonomically and functionally classified in order to study potential microbiome transcriptomic differences across environments. Taxonomic classification (indicative of transcriptionally active taxa) was performed on the unmapped reads (i.e., those that did not map the coral nor the photosymbiont genomes) with Kraken2 2.1.2[102], using the NCBI taxonomic information combined with the RefSeq libraries "bacteria", "archaea", "fungi", and "viral". Kraken-biom 1.0.1[103] was used to create a BIOM table from all Kraken reports that was then imported into R using the *import_biom* function in the phyloseq 1.40 R package[104]. Relative abundance of transcriptomic reads were plotted at the phylum level by site (following Morrow and coauthors[22]) using ggplot2 3.4.1[105], and a non-metric multidimensional scaling (NMDS) plot was performed with the *ordinate* and *plot_ordination* functions in phyloseq using the NMDS method on the Bray-Curtis dissimilarities among colonies (method = "NMDS", distance = "bray").

Differences in microbiome functionality were assessed between environments following Gardiner and coauthors[106]. Functional classification was performed with a tree-shaped hierarchy of GO terms using the Molecular Function subset[106]. This GO tree-shaped hierarchy was combined with the RefSeq libraries "bacteria", "archaea", "fungi" and "viral" to build a database and to perform a direct functional classification of our unmapped clean reads with Kraken2 2.1.2. Kraken-biom 1.0.1 was used to create a BIOM table from all Kraken reports that was then imported into R using the *import_biom* function in phyloseq. An NMDS plot was performed on Bray-Curtis dissimilarities among colonies as detailed above. Additionally, in order to identify Molecular Functions that significantly differed between environments, a Welch t-test was run for every GO term independently using the *oneway.test* function in the stats 4.2.3 R package[107], grouping samples by environment (seep vs control colonies). A Bonferroni correction was applied to the significance level to avoid false positives due to multiple testing.

**Statistics and reproducibility**. Four analyses were used to identify SNPs under natural selection or associated with pH in the coral host, and then those present in at least two of the four analyses were considered as under selection in order to avoid false positives. Subseqently, a Gene Set Enrichment Analysis was performed to detect over-represented GO terms in the candidate adaptive SNP annotations, using a false discovery rate approach to control for false positives. See more details in Methods section, subsection "Identification and annotation of loci under selection and genotype-environment association analyses in the coral host".

For the coral microbiome, a Welch t-test was run for every GO term independently using the *oneway.test* function in the stats 4.2.3 R package[107], in order to identify Molecular Functions that significantly differed between environments. The Welch t-test was chosen after checking for equal variances using the *var.test* in the stats 4.2.3 R package[107], which indicated that there were significant differences in variances and hence the classic t-test could not be applied. A Bonferroni correction was applied to the significance level to avoid false positives due to multiple testing, lowering the significance level from 0.05 to 0.00027.

**Reporting summary**. Further information on research design is available in the Nature Portfolio Reporting Summary linked to this article.

## Data availability

Data for this study was obtained from public repositories: NCBI SRA BioProject PRJNA362652[68], NCBI SRA BioProject PRJNA767661[71], and the reefgenomics.org[97] database. Numerical source data for graphs and charts can be found in Supplementary Data 3 and Supplementary Data 4. All other data are available from the corresponding author on reasonable request.

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

## Acknowledgements

We thank Jeffrey Centino and Dr. Bastian Bentlage for their computing support. This work was supported by the National Science Foundation NSF-EPSCoR Grant No. OIA-1946352. Funding also came from the Spanish Government grants ADAPTIVE (PGC2018-100735-B-I00/MCIU/AEI/FEDER, UE) and ENVIOME (PID2021-128094NB-I00/MCIN/AEI/10.13039/501100011033/ and FEDER una manera de hacer Europa), a "Ramón y Cajal" contract to R.P.-P. (RYC2018-025070-I), and the project "DIVERGEN—Ayudas fundación BBVA a Proyectos Investigación Científica 2021".

## Author contributions

C.L. and R.P.-P. conceived and designed the study; C.L. performed the analyses; C.L., R.P.-P. and S.L. interpreted and discussed the results; S.L. obtained funding; C.L. wrote the first version of the manuscript, and all coauthors contributed to the current version.

## Competing interests

The authors declare no competing interests.
