## [Peer Review File · Communications Biology]

Reviewers' comments:

Reviewer #1 (Remarks to the Author):

The authors present a timely and novel analysis of available genomic datasets to evaluate the adaptive mechanisms allowing coral populations to thrive in CO₂ seeps. It's very rewarding to see proper use of freely accessible data, justifying all the hard work researchers invest in curating and depositing data in repositories. Authors, utilize appropriate statistical and bioinformatic tools to identify signals of selection on host genes, genetic structure between symbiont populations, and functional shifts on microbiome communities as mechanisms of holobiont's local adaptation. The manuscript is concise, well written and shows an appropriate use of the literature. However, I have minor comments and suggestions that require review and clarification.

The use of migration in line 45 can be confusing and should be replaced by range expansion or other term that better describe the process in sessile organisms that cannot migrate per-se.

In line 98 the number of mapped reads seems very low compared to other RNA-seq studies, being even lower to what Kenkel et al. (2018) reported with the same dataset. How can this low mapping be explained? Could this affect the obtained results? Wouldn't this potentially bias SNP discovery?

In addition, Kenkel transcriptomic dataset was created with TagSeq. Substantial differences with RNA-seq should be clearly stated in the manuscript and used in the discussion of the results to clearly define the limitations of the conclusions. This was briefly mentioned in the methods section, but I consider it should be discussed further in the manuscript, especially considering the format of the manuscript.

As the authors state, tag-based RNAseq only produces short fragments of the transcriptome with many reads covering mostly untranslated regions of the gene. This has benefits, as described by the authors in line 51 of the methods section but imposes some technical issues for example to mapping to genomes (generally with poor UTR annotations) that were not discussed. Did the authors consider this in the analysis? For example, biases related with SNPs being detected only in portions of genes (maybe only untranslated regions with no functional influence)?

In line 200: "Signals of environmental-driven adaptation were also found in eight additional differentially regulated loci from Kenkel and coauthors (2018), suggesting cis-regulation of gene expression in key components of the coral response to acidification, including calcification, transcription processes, and the immune system", and also in Table 1 "genes with signals of cis-regulation" are inferred from Kenkel et al 2018 datasets. What do you mean with cis-regulation? Were regulatory elements identified in the dataset?

Reviewer #2 (Remarks to the Author):

The manuscript "Synergistic genomic mechanisms of adaptation to ocean acidification in a coral holobiont" aims to identify adaptive signals in the *Acropora millepora* coral holobiont at four different sites along a natural CO₂ seep using a previously published transcriptomic dataset. The authors identify high levels of population connectivity across the different sites, however, are able to identify potential adaptive genes associated with coral calcification and response to ocean acidification in samples obtained from low pH sites. Overall, this is an interesting study that leverages existing data

to address potential mechanistic pathways of coral adaptation under high-CO₂ scenarios. While the evidence of potential adaptive genes that allow for coral host adaptation are well-presented here, I am not as convinced at present by the authors conclusion that different symbiont genetic clusters exist across the pH gradient. The authors need to provide more compelling evidence for the photosymbiont role in the holobiont adaptive response or downplay this component of the manuscript. Similarly, I found the microbiome portion of this study interesting, but it is again presented in a way that could use more support throughout to convince readers of the role of the microbiome in holobiont response to low pH conditions. With some updates, I believe this will be a valuable contribution to the field of coral biology/physiology and will be of great interest to the field. Please refer to specific comments below.

Specific Comments:

L28: comma after "years"

L32-33: suggest just saying "carbonate" instead of "carbonate minerals"

L37: same as above

L35-42: I recommend reviewing Global declines in coral reef calcium carbonate production under ocean acidification and warming by Cornwall et al. regarding the future of coral reefs under ocean acidification and warming.

L98: write out what SNP stands for when using it first time

L106-110: Somewhere here (or earlier in the manuscript) the authors should introduce their site locations, average site conditions, and maybe even include a map to help guide readers with this portion of their work.

L239-242: I am not entirely convinced by this statement based solely on your presented heatmap. I strongly recommend adding some additional analyses/visualizations of your photosymbiont data to better support these differences across pH gradients.

Figure 2 – Please include the abbreviations that you use on the figure itself in the caption when you mention the site names. It may also benefit your readers to use different shapes in the DAPC along with different colours for interpretation of the data. The orange and red are also a bit tough to distinguish from one another in panel B so please consider modifying your colour choices here.

Figure 3 – I really like this method of visualizing your GO terms in your study. It looks like the line styles are different in A/B from C. Consider using a darker font for the REVIGO terms in the light greens and yellow to improve readability. Also consider making your BP, MF, and CC category labels stick out more so readers can clearly see which category the terms are associated with.

Figure 4 – Again, your red and orange appear to be pretty similar to one another. Considering selecting new colours. Please specify in your caption what the clusters on the right of the heat map refer to.

Methods

Make sure you are citing the R packages and software you use appropriately

L5-15: Some of these details should be provided earlier in the manuscript (as mentioned above) to provide context of the spatial and environmental gradient being assessed.

L44-45: make sure you are also citing the software here as appropriate

L109: please include the number of PCs and Das retained in each of your DAPCs

L110: please cite R package.

Reviewer #3 (Remarks to the Author):

This paper claims to have identified synergistic mechanisms of adaptation to ocean acidification in a coral holobiont based on a re-analysis of published transcriptomic, genomic, and metagenomic data from corals at and near a CO₂ seep.

Unfortunately, this approach simply cannot provide information about the question of interest for a variety of reasons that include (but are not limited to):

1) GO-KEGG terms are not ideal for non-model species such as corals as they do not capture their unique biology. These terms are largely based on rodents, fly, *C. elegans*, etc which, among other things, do not produce a massive calcium carbonate exoskeleton nor they establish symbiosis with algae.

As a result, the analyses include genes such as bone morphometric protein 7, endothelin-converting enzyme homolog, and ependymin-like protein despite the fact that corals do not have bones, an endothelium, or cerebrospinal fluid.

2) mRNA levels do not necessarily reflect protein levels, which questions the functional relevance of the changes that were identified.

3) mRNA levels can be highly sensitive to local conditions at the time of sampling (e.g. factors that affect light levels such as depth and weather, food type and availability, time of the day, and a large list of etc). This means that the changes cannot be assumed to be due to the CO₂/pH condition.

4) Even if we assumed that mRNA levels did reflect protein levels, up- or down-regulation of a pathway would require concerted changes of the relevant proteins (not just one here and there) and in the same cells (not as an average throughout the colony).

5) Many enzymes are involved in multiple functions so their role cannot be ascribed to a single process. For example, carbonic anhydrase 2 is involved in both calcification and photosynthesis (and possibly many more functions). Is the analyses picking up a change related to one of them? To both? A combination whereby it is downregulated for one function and upregulated for the other? This applies to many genes/proteins.

6) Many enzymes are comprised of multiple subunits, and thus differential expression of a single subunit would not have any effect on function.

7) Transcriptomic analyses offer a correlation between a certain condition and mRNA levels. It cannot be used as evidence for cause-effect, especially when it refers to data obtained from corals from the field where multiple environmental and physiological parameters are simultaneously and continuously changing.

I realize the "transcriptomic approach" has become very popular and it is convenient. I also realize it requires a lot of work. However, this does not make it a suitable approach to study the question of interest.

Reviewers' comments:

Reviewer #1 (Remarks to the Author):

The authors present a timely and novel analysis of available genomic datasets to evaluate the adaptive mechanisms allowing coral populations to thrive in CO₂ seeps. It's very rewarding to see proper use of freely accessible data, justifying all the hard work researchers invest in curating and depositing data in repositories. Authors, utilize appropriate statistical and bioinformatic tools to identify signals of selection on host genes, genetic structure between symbiont populations, and functional shifts on microbiome communities as mechanisms of holobiont's local adaptation. The manuscript is concise, well written and shows an appropriate use of the literature. However, I have minor comments and suggestions that require review and clarification.

Response: We really appreciate this comment from Reviewer #1, as well as the rest of their suggestions that have helped to improve our manuscript.

The use of migration in line 45 can be confusing and should be replaced by range expansion or other term that better describe the process in sessile organisms that cannot migrate per se.

Response: Following Reviewer's suggestion we replaced "migration" by "poleward range expansions" in lines 49 and 51 (lines refer to the "TRACK_CHANGES" version of the manuscript).

In line 98 the number of mapped reads seems very low compared to other RNA-seq studies, being even lower to what Kenkel et al. (2018) reported with the same dataset. How can this low mapping be explained? Could this affect the obtained results? Wouldn't this potentially bias SNP discovery?

Response: Indeed, as Reviewer #1 mentions, the number of reads that we report is lower than the reported by Kenkel et al. (2018). This is due to the fact that our mapped reads values refer to the number of reads that pass our post-mapping filters, which include a

deduplication step. This deduplication step is highly recommended for SNP calling from RNAseq reads to mitigate biases due to PCR duplicates and, importantly, gene expression variation.

In the first version of the manuscript, this sentence seemed to suggest that our mapped reads values referred to the raw reads before the filtering steps, as Reviewer #1 noticed. In order to clarify this point, we changed this sentence in the new version of our manuscript: "A total of 40.7 million clean reads mapped the *Acropora millepora* reference genome and passed post-mapping filters, averaging 690,440 reads per individual, and 79,273 SNPs were obtained after variant calling and variant filtering."

In addition, Kenkel transcriptomic dataset was created with TagSeq. Substantial differences with RNA-seq should be clearly stated in the manuscript and used in the discussion of the results to clearly define the limitations of the conclusions. This was briefly mentioned in the methods section, but I consider it should be discussed further in the manuscript, especially considering the format of the manuscript.

Response: Following Reviewer's suggestion, we have expanded the discussion about the limitations of our study regarding the use of TagSeq, and have moved it from the methods to the discussion section, lines 282-292: "Unlike whole transcriptome sequencing, tag-based RNAseq only produces short fragments of the transcriptome, complementary to the 3'-end of the transcripts (Matz, 2018). Hence, our dataset could be missing other primordial gene regions bearing signals of adaptation that might have been detected with whole transcriptome or whole genome sequencing approaches. However, high linkage and low probability of recombination events within a single gene ensure that signals of selection found in tag-based RNAseq transcripts are representative of selection in that whole gene. In fact, because of the high chance of linkage within neighboring genomic regions, it is common practice in whole genome selection scans to produce thinned SNP datasets filtering physically linked SNPs within a few kbp, resulting in a similar exclusion of putative adaptive SNPs (e.g., Cooke et al., 2020; Mattingsdal et al., 2020)."

As the authors state, tag-based RNAseq only produces short fragments of the transcriptome with many reads covering mostly untranslated regions of the gene. This has benefits, as described by the authors in line 51 of the methods section but imposes some

technical issues for example to mapping to genomes (generally with poor UTR annotations) that were not discussed. Did the authors consider this in the analysis? For example, biases related with SNPs being detected only in portions of genes (maybe only untranslated regions with no functional influence)?

Response: As Reviewer #1 mentions, many of the tag-based RNAseq reads mapped 3'-UTR regions of the *Acropora millepora* genome, and many of them were not annotated. This is the reason why after annotating the SNP dataset with SnpEff (which only uses the actual genome annotations), we also ran BLASTN and BLASTX searches of the putative intergenic regions that some reads mapped to. Following Reviewer's comment, we specified these technical issues in the Methods section, lines 74-81: "As tag-based RNAseq produces fragments of the transcriptome near the 3'-end of the transcripts (Matz, 2018), many reads were expected to map 3'-UTR regions, which are generally poorly annotated in reference genomes. Indeed, 16.7% of the SNPs were annotated as intergenic SNPs by SnpEff using *A. millepora* genome annotations from Fuller et al. (2020). To improve annotations of these transcripts, BLASTN and BLASTX searches against the nr NCBI database (Altschul et al., 1997) were performed using default parameters."

As we stated in the previous response, we consider that due to the high linkage within a particular gene, adaptive signals in any part of a transcript indicate adaptation in that gene, following many studies that produce thinned SNP datasets as they keep the same information, do not include linked SNPs and are computationally easier to analyze.

In line 200: "Signals of environmental-driven adaptation were also found in eight additional differentially regulated loci from Kenkel and coauthors (2018), suggesting cis-regulation of gene expression in key components of the coral response to acidification, including calcification, transcription processes, and the immune system", and also in Table 1 "genes with signals of cis-regulation" are inferred from Kenkel et al 2018 datasets. What do you mean with cis-regulation? Were regulatory elements identified in the dataset?

Response: In this study, we didn't aim to identify regulatory regions in the dataset, instead, we checked whether any of the genes that we identified with adaptive SNPs were differentially expressed between control and seep colonies by Kenkel et al. (2018). By using

“signals of cis-regulation” in the genes shown in Table 1, we meant that those genes presented both signals of adaptation to the seep environment and differential expression between seep and control environments. Although we didn’t directly test for cis-eQTL (cis-expression quantitative trait loci), the fact that a gene is both adaptive and differentially expressed suggests cis-regulation of its gene expression.

As we didn’t explicitly test for cis-eQTL and the “cis-regulation” term might cause confusion, we changed Table 1 “Genes with signals of cis-regulation in *Acropora millepora*” to “Genes with signals of adaptation that were also differentially expressed between seep and control sites in Kenkel et al. (2018)”.

Reviewer #2 (Remarks to the Author):

The manuscript “Synergistic genomic mechanisms of adaptation to ocean acidification in a coral holobiont” aims to identify adaptive signals in the *Acropora millepora* coral holobiont at four different sites along a natural CO₂ seep using a previously published transcriptomic dataset. The authors identify high levels of population connectivity across the different sites, however, are able to identify potential adaptive genes associated with coral calcification and response to ocean acidification in samples obtained from low pH sites. Overall, this is an interesting study that leverages existing data to address potential mechanistic pathways of coral adaptation under high-CO₂ scenarios. While the evidence of potential adaptive genes that allow for coral host adaptation are well-presented here, I am not as convinced at present by the authors conclusion that different symbiont genetic clusters exist across the pH gradient. The authors need to provide more compelling evidence for the photosymbiont role in the holobiont adaptive response or downplay this component of the manuscript. Similarly, I found the microbiome portion of this study interesting, but it is again presented in a way that could use more support throughout to convince readers of the role of the microbiome in holobiont response to low pH conditions. With some updates, I believe this will be a valuable contribution to the field of coral biology/physiology and will be of great interest to the field. Please refer to specific comments below.

Response: We really appreciate these comments from Reviewer #2, which made us re-think

and re-analyze both the photosymbiont and the microbiome analyses. As Reviewer #2 mentioned, these parts were lacking support in the previous version of the manuscript, which we have improved and extensively revised in the newly submitted version. We summarize the main improvements for each part below:

Main improvements in the *Cladocopium goreau* population genetics analyses:

- We called SNPs with freebayes, adding the --pool-continuous flag to call SNPs from pooled samples. Compared to our previous approach, freebayes allows missing data, which we limited to SNPs present in at least 70% of the colonies, using the --max-missing 0.7 flag. We obtained 2,707 SNPs using this approach, a great improvement from the 214 SNPs previously obtained with poolfstat.
- We performed a DAPC analysis with this SNP dataset. We present this results in the new Figure 5, together with the pairwise F_{ST} heatmap for the new SNP dataset.
- As in the previous version of the manuscript, we also test two different models using AMOVAs in the new SNP dataset: a spatial model and an environmental model. The AMOVA results and tables have been updated in the new version of the manuscript and are presented in Table 2.
- These new results are discussed in the revised version of the manuscript. Briefly, we show a combined effect of both space and environment in the genetic structure of the photosymbionts. We show a spatial structuring effect in the DAPC (new Figure 5), with samples clustering by reef in the first DAPC axis (Dobu samples on the left, Upa-Upasina samples on the right). Then, samples from Dobu seep and control sites are differentiated along the second DAPC axis, and samples from Upa-Upasina seep and control sites are separated along the third DAPC axis. These results suggest a strong reef component in the *Cladocopium goreau* differentiation (along the first DAPC axis), but also highlight the environmental component: samples from Dobu, where the pH difference between seep and control is higher (7.72-8.01), are differentiated along the second DAPC axis, while samples from Upa-Upasina, where the pH difference is lower (7.81-7.98) are differentiated along the third DAPC axis. This was supported by the AMOVA results, showing that both spatial and

environmental models explain a significant part of the genetic variation, with the spatial model explaining a slightly higher percentage than the environmental model: between reefs 2.05%, between sites within reef 1.68%, between environments (controls vs seeps) 0.92%. Despite being low, the three percentages are highly significant (p -values <0.01).

Main improvements in the microbiome functionality analyses:

- First, we would like to highlight that, as mentioned in the manuscript, microbial community changes between *A. millepora* colonies from these very same sites were already studied and characterized in Morrow et al. (2015). They found significant differences in the microbial communities between seep and control sites using 16S amplicon sequencing, which is the preferred method to study microbial species abundances and changes in the species composition. On the other hand, the dataset we analyzed in our study is a transcriptomic dataset, revealing microbiome activity and functionality. These two methods inform about different features of the microbial community, that when taken together can be seen as complementary. Importantly, the two methods don't need to agree: for instance, rare species (low counts on 16S amplicon sequencing) could be highly transcriptionally active (high counts on a metatranscriptomic dataset), or vice versa.
- In our revised manuscript, we classified the microbiome transcriptomic reads in two different ways: taxonomically and functionally (previously, we only classified them functionally).
- Regarding the taxonomic classification (i.e., transcriptionally active taxa), we obtained similar proportions in all colonies, indicating similar abundances of metatranscriptomic reads from each taxonomic group regardless of site and environment. These results contrast with the different microbial communities found by Morrow et al. (2015), implying that although the species composition is different, each taxonomic group maintains a similar transcriptional activity across environments.
- Similarly, for the functional classification, we also obtained similar proportions in all colonies, revealed by the NMDS where the samples do not cluster by colony origin or environment. In order to reveal the particular molecular functions that differ between

control and seep colonies, we now performed a Welch t-test for each GO term independently in order to test for significant differences in metatranscriptomic reads between control and seep colonies. We consider that this is more appropriate than the simpler analysis that we performed in the previous version of the manuscript, as the simpler is a method used to analyze species abundances. With this new approach, we did not find any GO term with significantly different read counts between environments, in agreement with the microbial homogeneity shown by both NMDS and the similar phyla percentages in each site (see new Figure 6).

- These new results reveal that although microbial communities change between seep and control sites (Morrow et al., 2018), the taxonomic and functional relative abundance of metatranscriptomic reads are not significantly different. We further discuss this results in the discussion section lines 316-340.

Specific Comments:

L28: comma after "years"

Response: As suggested by Reviewer #2, comma was added after "years".

L32-33: suggest just saying "carbonate" instead of "carbonate minerals"

Response: "carbonate minerals" was changed to "carbonate".

L37: same as above

Response: "carbonate minerals" was changed to "carbonate".

L35-42: I recommend reviewing Global declines in coral reef calcium carbonate production under ocean acidification and warming by Cornwall et al. regarding the future of coral reefs under ocean acidification and warming.

Response: Following Reviewer's comment, we added the following sentence in lines 44-46: "Moreover, the combined effects of global ocean acidification and warming appear to increase global reductions in net carbonate production and accretion of most coral reefs (Cornwall et al., 2021)."

L98: write out what SNP stands for when using it first time

Response: "Single Nucleotide Polymorphisms (SNPs)" was added the first time the term was used, in line 86.

L106-110: Somewhere here (or earlier in the manuscript) the authors should introduce their site locations, average site conditions, and maybe even include a map to help guide readers with this portion of their work.

Response: Following Reviewer's recommendation, we introduced the study sites in the revised version of the manuscript, in the last paragraph of the introduction, lines 77-82: "This CO₂ seep system has been relatively well studied, with changes reported in diversity and composition of corals and reef-associated macroinvertebrate communities at Dobu and Upa-Upasina reefs (Fabricius et al., 2011; 2014). Seep and control sites have been identified at both Dobu and Upa-Upasina reefs, with pH ranging from 7.72 to 7.81 in the seep sites and from 7.98 to 8.01 in the control sites, respectively (see Methods and Fabricius et al., 2011 for a thorough description of the study site)."

Also, as suggested by Reviewer #2 we added a map in the new Figure 1.

L239-242: I am not entirely convinced by this statement based solely on your presented heatmap. I strongly recommend adding some additional analyses/visualizations of your photosymbiont data to better support these differences across pH gradients.

Response: As detailed above, following Reviewer #2 comments, new analyses and visualizations of the *Cladocopium goreau* dataset were added in the revised version of the manuscript. Methods lines 138-148. Results lines 161-175.

Figure 2 – Please include the abbreviations that you use on the figure itself in the caption when you mention the site names. It may also benefit your readers to use different shapes in the DAPC along with different colours for interpretation of the data. The orange and red are also a bit tough to distinguish from one another in panel B so please consider modifying your colour choices here.

Response: Abbreviations are now included in the figure captions. Following reviewer's comment on colors, we changed the colors used to represent sampling sites throughout the figures, in order to ease their interpretation: teal for Dobu reef, brown for Upa-Upasina reef; darker colors for control sites, lighter colors for seep sites. We chose these colors after checking at viz palette (<https://projects.susielu.com/viz-palette>), to make sure the colors are accessible to people with all types of color blindness.

Figure 3 – I really like this method of visualizing your GO terms in your study. It looks like the line styles are different in A/B from C. Consider using a darker font for the REVIGO terms in the light greens and yellow to improve readability. Also consider making your BP, MF, and CC category labels stick out more so readers can clearly see which category the terms are associated with.

Response: Many thanks for this comment and your suggestions to improve Figure readability and clarity. We checked and standardized line styles.

Instead of using a darker font for the REVIGO terms in the light greens and yellow as Reviewer #2 suggests, we kept the white for the front and made the greens and yellow darker. We hope this will help readability. We are happy to modify it if it's required by the editor.

We increased the font size of BP, MF and CC to make them stick out more as suggested by Reviewer #2.

Figure 4 – Again, your red and orange appear to be pretty similar to one another. Considering selecting new colours. Please specify in your caption what the clusters on the right of the heat map refer to.

Response: Colors were changed, see our response about on Figure 2 above.

Methods

Make sure you are citing the R packages and software you use appropriately

Response: All R packages and software citations were reviewed.

L5-15: Some of these details should be provided earlier in the manuscript (as mentioned above) to provide context of the spatial and environmental gradient being assessed.

Response: As stated above, a brief introduction to the study area is now included in the introduction (lines 79-82). We also added in the introduction a concise explanation of the methods that we used to test the three hypotheses (lines 86-107). Also, we added a new Figure 1 showing the schematic workflow of our study.

L44-45: make sure you are also citing the software here as appropriate

Response: We checked this and consider that our citation is correct, as we are citing the R package in which the function `rda` is contained (vegan package, citation: Dixon, 2003). Please let us know if we misinterpreted this comment and if you meant something else . Did you maybe mean the BayPass citation? We moved BayPass citation from line 44 to line 43, where BayPass is first mentioned.

L109: please include the number of PCs and DAs retained in each of your DAPCs

Response: Retained PC and DA axes are now included, thanks.

L110: please cite R package.

Response: We checked this and consider that our citation style is correct. As adegenet R package was already cited previously when it first appeared, we did not cite it again. We can change this if the citation style of the journal requires us to include the software/R package citation every time the software appears in the manuscript.

Reviewer #3 (Remarks to the Author):

This paper claims to have identified synergistic mechanisms of adaptation to ocean acidification in a coral holobiont based on a re-analysis of published transcriptomic, genomic, and metagenomic data from corals at and near a CO₂ seep. Unfortunately, this approach simply cannot provide information about the question of interest for a variety of reasons that include (but are not limited to):

1) GO-KEGG terms are not ideal for non-model species such as corals as they do not capture their unique biology. These terms are largely based on rodents, fly, *C. elegans*, etc which, among other things, do not produce a massive calcium carbonate exoskeleton nor they establish symbiosis with algae.

As a result, the analyses include genes such as bone morphometric protein 7, endothelin-converting enzyme homolog, and ependymin-like protein despite the fact that corals do not have bones, an endothelium, or cerebrospinal fluid.

Response: We respectfully disagree with reviewer #3's opinion. We could cite a never-ending list of papers that make use of GO terms or KEGG pathways on non-model species. In fact, GO term annotations are a great tool for non-model species for which there are no experiments linking specific genes to specific functions. This is based on the functional homology of orthologous genes across the Tree of Life. Please see Gabaldón & Koonin (2013) in Nature Reviews Genetics (<https://doi.org/10.1038/nrg3456>) and Fernández & Gabaldón (2020) in Nature Ecology & Evolution (<https://doi.org/10.1038/s41559-019-1069-x>).

However, luckily, there is a large body of literature that focuses on corals, and more specifically on experiments characterizing the genes that regulate coral calcification. That's why we are able to link genes with signals of local adaptation with their particular function,

as they were previously identified and tested in corals. In fact, sometimes they were even characterized in the species we worked on here, *Acropora millepora*, or other closely related *Acropora* species, as detailed in Table 1.

The fact that the names of the genes include words like “bone” or “endothelin” does not imply that corals have bones or endothelium, and is by no means related to the first statement of Reviewer #3 regarding GO terms and KEGG pathways. Genes are named when they are first discovered (indeed, mostly in model species such as humans, flies, rodents or *C. elegans*), and then, when a homologue of that particular gene is found in a non-model species, that same gene name is used, independently of whether the non-model species has bones or endothelium.

General response to comments 2 to 7: The following comments are all based on the assumption that our manuscript analyzes “mRNA levels/ gene expression levels”, which is incorrect. In this study, we mainly analyzed Single Nucleotide Polymorphisms (SNPs), which were called from a publicly available dataset of tag-based RNAseq reads. This specific approach is akin to using RADseq, GBS, whole exome sequencing or any other kind of reduced representation genome sequencing approach, except that it uses tagSeq reads as starting material instead. This novel approach actually allows us to address multiple questions of interest such as genomic signals of local adaptation in the coral host, population genetic differentiation in the photosymbionts, and microbiome metatranscriptomic characterization. At no point did we worked on mRNA levels, gene expression levels, protein levels or genes being up- or downregulated. Hence, all the comments below do not apply to our study.

However, following Reviewer #3’s comments and to avoid readers being similarly confused, we have now included a detailed schematic of our approaches and analyses in the context of Kenkel et al., 2018’s study: Please see new Figure 1.

2) mRNA levels do not necessarily reflect protein levels, which questions the functional relevance of the changes that were identified.

Response: This comment does not apply here since we did not investigate mRNA, gene expression or protein concentration but looked at point mutations (SNP) to identify potential genes under selection.

3) mRNA levels can be highly sensitive to local conditions at the time of sampling (e.g. factors that affect light levels such as depth and weather, food type and availability, time of the day, and a large list of etc). This means that the changes cannot be assumed to be due to the CO₂/pH condition.

Response: This comment does not apply here. We only looked at point mutations which are not sensitive to local conditions, food type, or time of sampling.

4) Even if we assumed that mRNA levels did reflect protein levels, up- or down-regulation of a pathway would require concerted changes of the relevant proteins (not just one here and there) and in the same cells (not as an average throughout the colony).

Response: This comment does not apply here since we do not analyze mRNA or protein concentrations or the down or up regulation of pathways.

5) Many enzymes are involved in multiple functions so their role cannot be ascribed to a single process. For example, carbonic anhydrase 2 is involved in both calcification and photosynthesis (and possibly many more functions). Is the analyses picking up a change related to one of them? To both? A combination whereby it is downregulated for one function and upregulated for the other? This applies to many genes/proteins.

Response: Again, our analyses do not investigate down or up-regulation of genes or enzymes (this was already done in the original paper Kenkel et al. 2018), so this comment does not apply here.

6) Many enzymes are comprised of multiple subunits, and thus differential expression of a single subunit would not have any effect on function.

Response: Same comment as above. We did not look into differential gene expressions but at genes under selection.

7) Transcriptomic analyses offer a correlation between a certain condition and mRNA levels. It cannot be used as evidence for cause-effect, especially when it refers to data obtained from corals from the field where multiple environmental and physiological parameters are simultaneously and continuously changing.

I realize the "transcriptomic approach" has become very popular and it is convenient. I also realize it requires a lot of work. However, this does not make it a suitable approach to study the question of interest.

Response: Respectfully, this is a misplaced comment and should maybe have been submitted to the original Kenkel et al., 2018 paper since unlike us, they used gene expression to make inferences on physiological differences between corals colony in contracting environments. We did not since our analyses only rely on actual genomic mutations.

REVIEWERS' COMMENTS:

Reviewer #1 (Remarks to the Author):

Thank you very much to the authors for thoroughly address my comments of the submitted manuscript. After reviewing the changes resulting from the comments of all referees, I consider the manuscript of high quality and propose its publication.

Reviewer #2 (Remarks to the Author):

After reviewing the response to reviewer comments and the revised manuscript, I am very pleased with the authors' efforts to address concerns and update the manuscript. They have carefully and clearly addressed the comments/concerns of the reviewers and I think their manuscript is improved from it. My only suggestion to the authors is that they strongly consider making their code and scripts publicly available (and link it in the manuscript) to further facilitate open access science that they were able to take advantage of themselves for this study. I look forward to this paper being published!

Reviewer #3 (Remarks to the Author):

I apologize that my previous review focused on coral host transcriptomics and did not offer comments about the SNPs analysis.

But while the analyses may "only rely on actual genomic mutations", the significance of the SNPs/genetic variants is exclusively based on conclusions from previous transcriptomics studies. Respectfully, it does not matter whether these concerns should have been raised during the peer-reviewing of Kenkel et al., 2018. What matters is whether the concerns are valid, and how they affect the interpretation of the SNPs results.

I hope that my comments do not come across as overly negative, I assure this is not my intention. My opinion is driven by my training as an experimentalist who values cause & effect, so I cannot help a certain level of skepticism about correlational studies such as this one. As a compromise, I suggest that the authors briefly address the points I raise below, and provide a tentative mechanistic explanation about how the identified SNPs may contribute to adaptation to high Co₂/low pH. I also suggest modifying the title to reflect the tentative and correlative nature of the results and conclusions (Maybe something along the lines of "Holobiont genomic signatures from corals living near volcanic CO₂ seeps")

Best regards.

Specific comments

-the genes that are presumably selected in response to high CO₂ (e.g. Table 1) are a compendium of mRNA responses, which suffer from the limitations listed in my original review: little-to-no knowledge about their function in corals, mismatches with protein levels, no info about cellular localization, etc. This generates a vicious circle whereby genomic/transcriptomic studies are cited by subsequent similar studies, thus creating the (false) impression that the functions of genes/proteins are known. Clearly, this affects the significance of the SNPs.

-GO-KEGG pathways is a main approach used by bioinformaticians, who also write papers endorsing

their use, which in turn are peer-reviewed and approved by other bioinformaticians. However, this does not make it an adequate method for non-model species. Furthermore, most scientists who adopt this approach admit to these issues "off the record", but continue to use it because it is a convenient way to analyze their datasets. Unfortunately, many of the conclusions are bound to be incorrect, potentially misleading conservation and policy-making efforts.

I am not asking the authors to change their analyses, I just hope they can provide a brief note listing these limitations.

-There are a few mechanistic studies on coral molecular biology (e.g. from groups at Monaco, Rutgers, Scripps, U. Haifa, KAUST, Carnegie) that have functionally identified proteins involved in coral calcification and symbiosis. These include CARPs and several bicarbonate, proton, calcium and ammonia transporters (i.e. slc4, NHE, PMCA, L-type calcium channels, NCX, Rh, AMT). Are any of these in the list of potential adaptive genes? Can this be discussed in the manuscript, please?

-I would find it very helpful if the manuscript can provide a brief explanation about SNPs and their relationship with adaptation potential.

-Is there any functional evidence that the data presented for coral animal, Symbiodineaceae, and bacterial communities reflect an actual "synergism"? Or is the data purely correlational?

REVIEWERS' COMMENTS:

Reviewer #1 (Remarks to the Author):

Thank you very much to the authors for thoroughly address my comments of the submitted manuscript. After reviewing the changes resulting from the comments of all referees, I consider the manuscript of high quality and propose its publication.

Response: We really appreciate this comment from Reviewer #1.

Reviewer #2 (Remarks to the Author):

After reviewing the response to reviewer comments and the revised manuscript, I am very pleased with the authors' efforts to address concerns and update the manuscript. They have carefully and clearly addressed the comments/concerns of the reviewers and I think their manuscript is improved from it. My only suggestion to the authors is that they strongly consider making their code and scripts publicly available (and link it in the manuscript) to further facilitate open access science that they were able to take advantage of themselves for this study. I look forward to this paper being published!

Response: We thank Reviewer #2 for this comment. There is no newly generated code associated with our study, it's been all copied/adapted from other sources. However, we will consider adding a Supplementary Material with links to our sources in the final version of the manuscript if it's requested by the editor.

Reviewer #3 (Remarks to the Author):

I apologize that my previous review focused on coral host transcriptomics and did not offer comments about the SNPs analysis.

But while the analyses may "only rely on actual genomic mutations", the significance of the SNPs/genetic variants is exclusively based on conclusions from previous transcriptomics studies. Respectfully, it does not matter whether these concerns should have been raised during the peer-reviewing of Kenkel et al., 2018. What matters is

whether the concerns are valid, and how they affect the interpretation of the SNPs results.

I hope that my comments do not come across as overly negative, I assure this is not my intention. My opinion is driven by my training as an experimentalist who values cause & effect, so I cannot help a certain level of skepticism about correlational studies such as this one. As a compromise, I suggest that the authors briefly address the points I raise below, and provide a tentative mechanistic explanation about how the identified SNPs may contribute to adaptation to high Co₂/low pH.

I also suggest modifying the title to reflect the tentative and correlative nature of the results and conclusions (Maybe something along the lines of "Holobiont genomic signatures from corals living near volcanic CO₂ seeps")

Best regards.

Specific comments

-the genes that are presumably selected in response to high CO₂ (e.g. Table 1) are a compendium of mRNA responses, which suffer from the limitations listed in my original review: little-to-no knowledge about their function in corals, mismatches with protein levels, no info about cellular localization, etc. This generates a vicious circle whereby genomic/transcriptomic studies are cited by subsequent similar studies, thus creating the (false) impression that the functions of genes/proteins are known. Clearly, this affects the significance of the SNPs.

Response: Following Reviewer's comment, we have clarified this issue in lines 421-426.

-GO-KEGG pathways is a main approach used by bioinformaticians, who also write papers endorsing their use, which in turn are peer-reviewed and approved by other bioinformaticians. However, this does not make it an adequate method for non-model species. Furthermore, most scientists who adopt this approach admit to these issues "off the record", but continue to use it because it is a convenient way to analyze their

datasets. Unfortunately, many of the conclusions are bound to be incorrect, potentially misleading conservation and policy-making efforts.

I am not asking the authors to change their analyses, I just hope they can provide a brief note listing these limitations.

Response: Following Reviewer's comment, we have clarified the use of GO terms in lines 438-431.

-There are a few mechanistic studies on coral molecular biology (e.g. from groups at Monaco, Rutgers, Scripps, U. Haifa, KAUST, Carnegie) that have functionally identified proteins involved in coral calcification and symbiosis. These include CARPs and several bicarbonate, proton, calcium and ammonia transporters (i.e. slc4, NHE, PMCA, L-type calcium channels, NCX, Rh, AMT). Are any of these in the list of potential adaptive genes? Can this be discussed in the manuscript, please?

Response: Unfortunately, none of the proteins listed by Reviewer #3 were found in our list of candidate adaptive SNPs. To ensure the manuscript remains concise and focused we prefer not to discuss protein functions that our analyses did not recover. Instead, we focus our discussion on the many others protein and gene functions that have also been identified as being involved in coral calcification and symbiosis (Table 1).

-I would find it very helpful if the manuscript can provide a brief explanation about SNPs and their relationship with adaptation potential.

Response: Following Reviewer's comment, we added a brief explanation in the Introduction section, lines 87-90.

-Is there any functional evidence that the data presented for coral animal, Symbiodineaceae, and bacterial communities reflect an actual "synergism"? Or is the data purely correlational?

Response: The inherent concept of a holobiont emphasizes the interdependence and synergistic nature of the host-photosymbiont-microbiome relationship. The concept of

holobiont also recognizes that these symbiotic communities are an integral part of the host organism's biology and can influence its development, physiology, and overall fitness. A coral colony is a holobiont and as such is considered to involve synergistic relationships. For example, our results show clear signs of selection in the coral genes involved in the establishment and regulation of the Symbiodiniaceae community, which in turn we find is significantly different between environments. For all these reasons we argue that the title of our manuscript is adequate, and we respectfully decide not to modify it.